# META BACK-TRANSLATION

## ABSTRACT

Back-translation (Sennrich et al., 2016) is an effective strategy to improve the performance of Neural Machine Translation (NMT) by generating pseudo-parallel data. However, several recent works have found that better translation quality of the pseudo-parallel data does not necessarily lead to a better final translation model, while lower-quality but diverse data often yields stronger results instead. In this paper we propose a new way to generate pseudo-parallel data for back-translation that directly optimizes the final model performance. Specifically, we propose a meta-learning framework where the back-translation model learns to match the forward-translation model's gradients on the development data with those on the pseudo-parallel data. In our evaluations in both the standard datasets WMT En-De'14 and WMT En-Fr'14, as well as a multilingual translation setting, our method leads to significant improvements over strong baselines.

## 1   INTRODUCTION

While Neural Machine Translation (NMT) delivers state-of-the-art performance across many translation tasks, this performance is usually contingent on the existence of large amounts of training data (Sutskever et al., 2014; Vaswani et al., 2017). Since large parallel training datasets are often unavailable for many languages and domains, various methods have been developed to leverage abundant monolingual corpora (Gulcehre et al., 2015; Cheng et al., 2016; Sennrich et al., 2016; Xia et al., 2016; Hoang et al., 2018; Song et al., 2019; He et al., 2020). Among such methods, one particularly popular approach is *back-translation* (BT; Sennrich et al. (2016)).

In BT, in order to train a source-to-target translation model, i.e., the *forward* model, one first trains a target-to-source translation model, i.e., the *backward* model. This backward model is then employed to translate monolingual data from the target language into the source language, resulting in a pseudo-parallel corpus. This pseudo-parallel corpus is then combined with the real parallel corpus to train the final forward translation model. While the resulting forward model from BT typically enjoys a significant boost in translation quality, we identify that BT inherently carries two weaknesses.

First, while the backward model provides a natural way to utilize monolingual data in the target language, the backward model itself is still trained on the parallel corpus. This means that the backward model's quality is as limited as that of a forward model trained in the vanilla setting. Hoang et al. (2018) proposed iterative BT to avoid this weakness, but this technique requires multiple rounds of retraining models in both directions which are slow and expensive.

Second, we do not understand how the pseudo-parallel data translated by the backward model affects the forward model's performance. For example, Edunov et al. (2018) has observed that pseudo-parallel data generated by sampling or by beam-searching with noise from the backward model train better forward models, even though these generating methods typically result in lower BLEU scores compared to standard beam search. While Edunov et al. (2018) associated their observation to the diversity of the generated pseudo-parallel data, diversity alone is obviously insufficient – some degree of quality is necessary as well.

In summary, while BT is an important technique, training a good backward model for BT is either hard or slow and expensive, and even if we have a good backward model, there is no single recipe how to use it to train a good forward model.

In this paper, we propose a novel technique to alleviate both aforementioned weaknesses of BT. Unlike vanilla BT, which keeps the trained backward model fixed and merely uses it to generate pseudo-

**Figure 1:** An example training step of meta back-translation to train a forward model translating English (En) into German (De). The step consists of two phases, illustrated from left to right in the figure. **Phase 1:** a backward model translates a De sentence taken from a monolingual corpus into a pseudo En sentence, and the forward model updates its parameters by back-propagating from canonical training losses on the pair (pseudo En, mono De). **Phase 2:** the updated forward model computes a cross-entropy loss on a pair of ground truth sentences (real En, real De). As annotated with the red path in the figure, this cross-entropy loss *depends* on the backward model, and hence can be back-propagated to update the backward model. Best viewed in colors.

parallel data to train the forward model, we continue to update the backward model throughout the forward model's training. Specifically, we update the backward model to improve the forward model's performance on a held-out set of ground truth parallel data. We provide an illustrative example of our method in Fig. 1, where we highlight how the forward model's held-out set performance depends on the pseudo-parallel data sampled from the backward model. This dependency allows us to mathematically derive an end-to-end update rule to continue training the backward model throughout the forward model's training. As our derivation technique is similar to meta-learning (Schmidhuber, 1992; Finn et al., 2017), we name our method *Meta Back-Translation* (MetaBT).

In theory, MetaBT effectively resolves both aforementioned weaknesses of vanilla BT. First, the backward model continues its training based on its own generated pseudo-parallel data, and hence is no longer limited to the available parallel data. Furthermore, MetaBT only trains one backward model and then trains one pair of forward model and backward model, eschewing the expense of multiple iterations in Iterative BT (Hoang et al., 2018). Second, since MetaBT updates its backward model in an end-to-end manner based on the forward model's performance on a held-out set, MetaBT no longer needs to explicitly understand the effect of its generated pseudo-parallel data on the forward model's quality.

Our empirical experiments verify the theoretical advantages of MetaBT with definitive improvements over strong BT baselines on various settings. In particular, on the classical benchmark of WMT En-De 2014, MetaBT leads to +1.66 BLEU score over sampling-based BT. Additionally, we discover that MetaBT allows us to extend the initial parallel training set of the backward model by including parallel data from slightly different languages. Since MetaBT continues to refine the backward model, the negative effect of language discrepancy is eventually rebated throughout the forward model's training, boosting up to +1.20 BLEU score for low-resource translation tasks.

## 2  A PROBABILISTIC PERSPECTIVE OF BACK-TRANSLATION

To facilitate the discussion of MetaBT, we introduce a probabilistic framework to interpret BT. Our framework helps to analyze the advantages and disadvantages of a few methods to generate pseudo-parallel data such as sampling, beam-searching, and beam-searching with noise (Sennrich et al., 2016; Edunov et al., 2018). Analyses of these generating methods within our framework also motivates MetaBT and further allows us to mathematically derive MetaBT's update rules in § 3.

**Our Probabilistic Framework.**  We treat a language $S$ as a probability distribution over all possible sequences of tokens. Formally, we denote by $P_S(\mathbf{x})$ the distribution of a random variable $\mathbf{x}$, whose each instance $x$ is a sequence of tokens. To translate from a source language $S$ into a target language $T$, we learn the conditional distribution $P_{S,T}(\mathbf{y}|\mathbf{x})$ for sentences from the languages $S$ and $T$ with a parameterized probabilistic model $P(\mathbf{y}|\mathbf{x}; \theta)$. Ideally, we learn $\theta$ by minimizing the objective:

$$J(\theta) = \mathbb{E}_{x,y \sim P_{S,T}(\mathbf{x},\mathbf{y})}[\ell(x,y;\theta)] \quad \text{where} \quad \ell(x,y;\theta) = -\log P(y|x;\theta) \qquad (1)$$

Since $P_{S,T}(x,y) = P_{S,T}(y)P_{S,T}(x|y) = P_T(y)P_{S,T}(x|y)$, we can refactor $J(\theta)$ from Eq. 1 as:

$$J(\theta) = \mathbb{E}_{y \sim P_T(\mathbf{y})} \mathbb{E}_{x \sim P_{S,T}(\mathbf{x}|y)}[\ell(x,y;\theta)] \qquad (2)$$

**Motivating BT.**  In BT, since it is not feasible to draw exact samples $y \sim P_T(\mathbf{y})$ and $x \sim P_{S,T}(\mathbf{x}|y)$, we rely on two approximations. First, instead of sampling $y \sim P_T(\mathbf{y})$, we collect a corpus $D_T$ of

monolingual data in the target language $T$ and draw the samples $y \sim \text{Uniform}(D_T)$. Second, instead of sampling $x \sim P_{S,T}(\mathbf{x}|y)$, we *derive* an approximate distribution $\widehat{P}(\mathbf{x}|y)$ and sample $x \sim \widehat{P}(\mathbf{x}|y)$. Before we explain the derivation of $\widehat{P}(\mathbf{x}|y)$, let us state that with these approximations, the objective $J(\theta)$ from Eq. 2 becomes the BT following objective:

$$\widehat{J}_{\text{BT}}(\theta) = \mathbb{E}_{y \sim \text{Uniform}(D_T)} \mathbb{E}_{x \sim \widehat{P}(\mathbf{x}|y)} [\ell(x, y; \theta)] \tag{3}$$

Rather unsurprisingly, $\widehat{P}(\mathbf{x}|y)$ in Eq. 3 above is derived from a pre-trained parameterized backward translation model $P(\mathbf{x}|\mathbf{y}; \psi)$. For example:

- $\widehat{P}(x|y) \triangleq \mathbf{1}[x = \text{argmax}_{\dot{x}} P(\dot{x}|y; \psi)]$ results in BT via beam-search (Sennrich et al., 2016).
- $\widehat{P}(x|y) \triangleq P(x|y; \psi)$ results in BT via sampling (Edunov et al., 2018).
- $\widehat{P}(x|y) \triangleq \mathbf{1}[x = \text{argmax}_{\dot{x}} \widetilde{P}(\dot{x}|y; \psi)]$ results in BT via noisy beam-search (Edunov et al., 2018) where $\widetilde{P}(\mathbf{x}|y; \psi)$ denotes the joint distribution of the backward model $P(\mathbf{x}|y; \psi)$ and the noise.

Therefore, we have shown that in our probabilistic framework for BT, three common techniques to generate pseudo-parallel data from a pre-trained backward model correspond to different derivations from the backward model's distribution $P(\mathbf{x}|\mathbf{y}; \psi)$. Our framework naturally motivates two questions: (1) given a translation task, how do we tell which derivation of $\widehat{P}(\mathbf{x}|y)$ from $P(\mathbf{x}|\mathbf{y}, \psi)$ is better than another? and (2) can we derive better choices for $\widehat{P}(\mathbf{x}|y)$ from a pre-trained backward model $P(\mathbf{x}|\mathbf{y}; \psi)$ according to the answer of question (1)?

**Metric for the Generating Methods.** In the existing literature, the answer for our first question is relatively straightforward. Since most papers view the method of generating pseudo-parallel data as a hyper-level design, i.e. similar to the choice of an architecture like Transformer or LSTM, and hence practitioners choose one method over another based on the performance of the resulting forward model on held-out validation sets.

**Automatically Derive Good Generating Methods.** We now turn to the second question that our probabilistic framework motivates. Thanks to the generality of our framework, *every* choice for $\widehat{P}(\mathbf{x}|y)$ results in an optimization objective. Using this objective, we can train a forward model and measure its validation performance to evaluate our choice of $\widehat{P}(\mathbf{x}|y)$. This process of choosing and evaluating $\widehat{P}(\mathbf{x}|y)$ can be posed as the following bi-level optimization problem:

$$\text{Outer loop:} \quad \widehat{P}^* = \underset{\widehat{P}}{\text{argmax}} \, \text{ValidPerformance}(\theta^*_{\widehat{P}}),$$

$$\text{Inner loop:} \quad \theta^*_{\widehat{P}} = \underset{\theta}{\text{argmin}} \, \widehat{J}_{\text{BT}}(\theta; \widehat{P}), \tag{4}$$

$$\text{where} \quad \widehat{J}_{\text{BT}}(\theta; \widehat{P}) = \mathbb{E}_{y \sim \text{Uniform}(D_T)} \mathbb{E}_{x \sim \widehat{P}(\mathbf{x}|y)} [\ell(x, y; \theta)]$$

The optimal solution of this bi-level optimization problem can potentially train a forward model that generalizes well, as the forward model learns on a pseudo-parallel dataset and yet achieves a good performance on a held-out validation set. Unfortunately, directly solving this optimization problem is not feasible. Not only is the inner loop quite expensive as it includes training a forward model from scratch according to $\widehat{P}$, the outer loop is also poorly defined as we do not have any restriction on the space that $\widehat{P}$ can take. Next, in § 3, we introduce a restriction on the space that $\widehat{P}$ can take, and show that our restriction turns the task of choosing $\widehat{P}$ into a differentiable problem which can be solved with gradient descent.

## 3 META BACK-TRANSLATION

Continuing our discussion from § 2, we design *Meta Back-Translation* (MetaBT) which finds a strategy to generate pseudo-parallel data from a pre-trained backward model such that if a forward model training on the generated pseudo-parallel data, it will achieve a strong performance on a held-out validation set.

**The Usage of "Validation" Data.** Throughout this section, readers will see that MetaBT makes extensive use of the "validation" set to provide feedback for refine the pseudo-parallel data's generating strategy. Thus, to avoid nullifying the meaning of a held-out validation set, we henceforth refer to the ground-truth parallel dataset where the forward model's performance is measured throughout its training as the *meta validation dataset* and denote it by $D_{\text{MetaDev}}$. Other than this meta validation set, we also have a separate validation set for hyper-parameter tuning and model selection.

**A Differentiable Bi-level Optimization Problem.** We now discuss MetaBT, starting with formulating a differentiable version of Problem 4. Suppose we have pre-trained a paramterized backward translation model $P(\mathbf{x}|\mathbf{y}; \psi)$. Instead of designing the generating distribution $\widehat{P}(\mathbf{x}|\mathbf{y})$ by applying actions such as sampling or beam-search to $P(\mathbf{x}|\mathbf{y}; \psi)$, we let $\widehat{P}(\mathbf{x}|\mathbf{y}) \triangleq P(\mathbf{x}|\mathbf{y}; \psi)$ and continue to update the backmodel's parameters $\psi$ throughout the course of training the forward model. Clearly, under this association $\widehat{P}(\mathbf{x}|\mathbf{y}) \triangleq P(\mathbf{x}|\mathbf{y}; \psi)$, the parameters $\psi$ controls the generating distribution of the pseudo-parallel data to train the forward model. By setting the differentiable parameters $\psi$ as the optimization variable for the outer loop, we turn the intractable Problem 4 into a differentiable one:

$$\text{Outer loop:} \quad \psi^* = \operatorname*{argmax}_{\psi} \text{Performance}(\theta^*(\psi), D_{\text{MetaDev}})$$

$$\text{Inner loop:} \quad \theta^*(\psi) = \operatorname*{argmin}_{\theta} \mathbb{E}_{y \sim \text{Uniform}(D_T)} \mathbb{E}_{x \sim \widehat{P}(\mathbf{x}|y)}[\ell(x, y; \theta)] \tag{5}$$

Bi-level optimization problems whose both outer and inner loops operate on differentiable variables like Problem 5 have appeared repeatedly in the recent literature of meta-learning, spanning many areas such as learning initialization (Finn et al., 2017), learning hyper-parameters (Baydin et al., 2018), designing architectures (Liu et al., 2019), and reweighting examples (Wang et al., 2019b). We thus follow their successful techniques and design a two-phase alternative update rule for the forward model's parameters $\theta$ in the inner loop and the backward model's parameters $\psi$ in the outer loop:

**Phase 1: Update the Forward Parameters $\theta$.** Given a batch of monolingual target data $y \sim$ Uniform$(D_T)$, we sample the pseudo-parallel data $(\widehat{x} \sim P(\mathbf{x}|y; \psi), y)$ and update $\theta$ as if $(\widehat{x}, y)$ was real data. For simplicity, assuming that $\theta$ is updated using gradient descent on $(\widehat{x}, y)$, using a learning rate $\eta_\theta$, then we have:

$$\theta_t = \theta_{t-1} - \eta_\theta \nabla_\theta \ell(\widehat{x}, y; \theta) \tag{6}$$

**Phase 2: Update the Backward Parameters $\psi$.** Note that Eq. 6 means that $\theta_t$ depends on $\psi$, because $\widehat{x}$ is sampled from a distribution parameterized by $\psi$. This dependency allows us to compute the meta validation loss of the forward model at $\theta_t$, which we denote by $J(\theta_t(\psi), D_{\text{MetaDev}})$, and back-propagate this loss to compute the gradient $\nabla_\psi J(\theta_t(\psi), D_{\text{MetaDev}})$. Once we have this gradient, we can perform a gradient-based update on the backward parameter $\psi$ with learning rate $\eta_\psi$:

$$\psi_t = \psi_{t-1} - \eta_\psi \nabla_\psi \nabla_\theta J(\theta_t(\psi), D_{\text{MetaDev}}) \tag{7}$$

**Computing $\nabla_\psi J(\theta_t(\psi), D_{\text{MetaDev}})$.** Our derivation of this gradient utilizes two techniques: (1) the chain rule to differentiate $J(\theta_t(\psi), D_{\text{MetaDev}})$ with respect to $\psi$ via $\theta_t$; and (2) the log-gradient trick from reinforcement learning literature (Williams, 1992) to propagate gradients through the sampling of pseudo-source $\widehat{x}$. We refer readers to § A.1 for the full derivation. Here, we present the final result:

$$\nabla_\psi J(\theta_t(\psi), D_{\text{MetaDev}}) \approx - \left[ \nabla_\theta J(\theta_t, D_{\text{MetaDev}})^\top \cdot \nabla_\theta \ell(\widehat{x}, y; \theta_{t-1}) \right] \cdot \nabla_\psi \log P(\widehat{x}|y; \psi) \tag{8}$$

In our implementation, we leverage the recent advances in high-order AutoGrad tools to efficiently compute the gradient dot-product term via Jacobian-vector products. By alternating the update rules in Eq. 6 and Eq. 7, we have the complete MetaBT algorithm.

**Remark: An Alternative Interpretation of MetaBT.** The update rule of the backward model in Eq. 8 strongly resembles the REINFORCE equation from the reinforcement learning literature. This similarity suggests that the backward model is trained as if it were an agent in reinforcement learning. From this perspective, the backward model is trained so that the pseudo-parallel data sampled from it would maximize the "reward":

$$R(\widehat{x}) = \nabla_\theta J(\theta_t, D_{\text{MetaDev}})^\top \cdot \nabla_\theta \ell(\widehat{x}, y; \theta_{t-1}) \tag{9}$$

Since this dot-product measures the similarity in directions of the two gradients, it can be interpreted that MetaBT optimizes the backward model so that the forward model's gradient on pseudo-parallel

data sampled from the backward model is similar to the forward model's gradient *computed on the meta validation set.* This is a desirable goal because the reward guides the backward model's parameters to favor samples that are similar to those in the meta validation set.

## 4 A MULLTILINGUAL APPLICATION OF METABT

We find that the previous interpretation of MetaBT in Section 3 leads to a rather unexpected application MetaBT. Specifically, we consider the situation where the language pair of interest $S$-$T$ has very limited parallel training data. In such a situation, BT approaches all suffer from a serious disadvantage: since the backward model needs to be trained on the parallel data $T$-$S$, when the amount of parallel data is small, the resulting backward model has very low quality. The pseudo-parallel corpus generated from the low-quality backward model can contaminate the training signals of the forward model (Currey et al., 2017).

To compensate for the lack of initial parallel data to train the backward model, we propose to use parallel data from a related language $S'$-$T$ for which we can collect substantially more data. Specifically, we train the backward model on the union of parallel data $T$-$S'$ and $T$-$S$, instead of only $T$-$S$. Since this procedure results in a substantially larger set of parallel training data, the obtained backward model has a higher quality. However, since the extra $S'$-$T$ parallel data dominates the training set of the backward model, the pseudo source sentences sampled from the resulting backward model would have more features of the related language $S'$, rather than our language of interest $S$.

In principle, MetaBT can fix this discrepancy by adapting the backward model using the forward model's gradient on the meta validation set that only contains parallel data for $S$-$T$. This would move the back-translated pseudo source sentences closer to our language of interest $S$.

## 5 EXPERIMENTS

We evaluate MetaBT in two settings: (1) a standard back-translation setting to verify that MetaBT can create more effective training data for the forward model, and (2) a multilingual NMT setting to confirm that MetaBT is also effective when the backward model is pre-trained on a related language pair as discussed in § 4.

### 5.1 DATASET AND PREPROCESSING

**Standard**  For the standard setting, we consider two large datasets: WMT En-De 2014 and WMT En-Fr 2014[1], tokenized with SentencePiece (Kudo & Richardson, 2018) using a joint vocabulary size of 32K for each dataset. We filter all training datasets, keeping only sentence pairs where both source and target have no more than 200 tokenized subwords, resulting in a parallel training corpus of 4.5M sentence pairs for WMT En-De and 40.8M sentences for WMT En-Fr. For the target monolingual data, we collect 250M sentences in German and 61 million sentences in French, both from the WMT news datasets between 2007 and 2017. After de-duplication, we filter out the sentences that have more than 200 subwords, resulting in 220M German sentences and 60M French sentences.

**Multilingual**  The multilingual setting uses the multilingual TED talk dataset (Qi et al., 2018), which contains parallel data from 58 languages to English. We focus on translating 4 low-resource languages to English: Azerbaijani (az), Belarusian (be), Glacian (gl), Slovak (sk). Each low-resource language is paired with a corresponding related high-resource language: Turkish (tr), Russian (ru), Portuguese (pt), Czech (cs). We following the setting from prior work (Neubig & Hu, 2018; Wang et al., 2019a) and use SentencePiece with a separate vocabulary of 8K for each language.

### 5.2 BASELINES

Our first baseline is **No BT**, where we train all systems using parallel data only. For the standard setting, we simply train the NMT model on the WMT parallel data. For the multilingual setting, we train the model on the concatenation of the parallel training data from both the low-resource language

---
[1]Data link: http://www.statmt.org/wmt14/

and the high-resource language. The No BT baseline helps to verify the correctness of our model implementations. For the BT baselines, we consider two strong candidates:

- **MLE**: we sample the pseudo source sentences from a fixed backward model trained with MLE. This baseline is the same with sampling-based BT (Edunov et al., 2018). We choose sampling instead of beam-search and beam-search with noise as Edunov et al. (2018) found sampling to be stronger than beam-search and on par with noisy beam-search. Our data usage, as specified in § 5.1, is also the same with Edunov et al. (2018) on WMT. We call this baseline MLE to signify the fact that the backward model is trained with MLE and then is kept fixed throughout the course of the forward model's learning.
- **DualNMT** (Xia et al., 2016): this baseline further improves the quality of the backward model using reinforcement learning with a reward that combines the language model score and the reconstruction score from the forward model.

Note that for the multilingual setting, we use top-10 sampling, which we find has better performance than sampling from the whole vocabulary in the preliminary experiments.

## 5.3 IMPLEMENTATION

We use the Transformer-Base architecture (Vaswani et al., 2017) for all forward and backward models in our experiments' NMT models. All hyper-parameters can be found in § A.2. We choose Transformer-Base instead of Transformer-Large because MetaBT requires storing in memory both the forward model and the backward model, as well as the two-step gradients for meta-learning, which together exceeds our 16G of accelerator memory when we try running Transformer-Large. We further discuss this in § 7.

For the *standard* setup, we pre-train the backward model on the WMT parallel corpora. In the meta-learning phases that we described in § 3, we initialize the parameters $\psi_0$ using this pre-trained checkpoint. From this checkpoint, at each training step, our forward model is updated using two sources of data: (1) a batch from the parallel training data, and (2) a batch of sentences from the monolingual data, and their source sentences are sampled by the backward model.

For the *multilingual* setup, we pre-train the backward model on the reverse direction of the parallel data from both the low-resource and the high-resource languages. From this checkpoint, at each meta-learning step, the forward model receives two sources of data: (1) a batch of the parallel data from the low-resource language. (2) a batch of the target English data from the high-resource language, which are fed into the BT model to sample the pseudo-source data.

## 5.4 RESULTS

| BT Model Objective | Multilingual | | | | Standard | |
|---|---|---|---|---|---|---|
| | az-en | be-en | gl-en | sk-en | en-de | en-fr |
| No BT | 11.50 | 17.00 | 28.44 | 28.19 | 26.49 | 38.56 |
| MLE (Edunov et al., 2018) | 11.30 | 17.40 | 29.10 | 28.70 | 28.73 | 39.77 |
| DualNMT (Xia et al., 2016) | 11.69 | 14.81 | 25.30 | 27.07 | 25.71 | − |
| Meta Back-Translation | **11.92**[*] | **18.10**[*] | **30.30**[*] | **29.00** | **30.39**[*] | **40.28**[*] |

**Table 1:** BLEU scores of MetaBT and of our baselines in the standard bilingual setting and the multilingual setting. ∗ indicates statistically significant improvements with $p < 0.001$. Our tests follow (Clark et al., 2011).

We report the BLEU scores (Papineni et al., 2002) for all models and settings in Tab. 1. From the table, we observe that the most consistent baseline is MLE, which significantly improves over the No BT baseline. Meanwhile, DualNMT's performance is much weaker, losing to MLE on all tasks except for az-en where its margin is only +0.19 BLEU compared to No BT. For WMT En-Fr, we even observe that DualNMT often results in numerical instability before reaching 34 BLEU score and thus we do not report the result. By comparing the baselines' performance, we can see that continuing to train the backward model to outperform MLE is a challenging mission.

Despite such challenge, MetaBT consistently outperforms all baselines in both settings. In particular, compared to the best baselines, MetaBT's gain is up to +1.20 BLEU in the low-resource multilingual

setting, and is +1.66 BLEU for WMT En-De 14. We remark that WMT En-De 14 is a relatively classical benchmark for NMT and that a gain of +1.66 BLEU on this benchmark is very significant. While the gain of MetaBT over MLE on WMT En-Fr is somewhat smaller (+0.51 BLEU), our statistical test shows that the gain is still significant. Therefore, our experimental results confirm the theoretical advantages of MetaBT. Next, in § 5.5, we investigate the behaviors of MetaBT to further understand how the method controls the generating process of pseudo-parallel data.

## 5.5 ANALYSIS

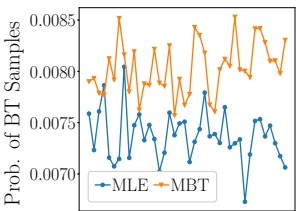

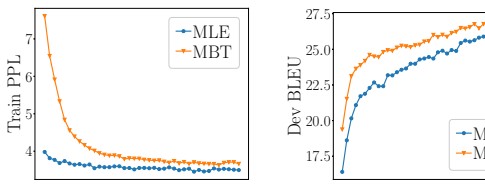

**Figure 2:** Probability of pseudo-parallel from the *forward* model for WMT'14 En-Fr. MetaBT produces less diverse data to fit the model better.

**Figure 3:** Training PPL and Validation BLEU for WMT En-De throughout the forward model's training. MetaBT leads to consistently higher validation BLEU by generating pseudo-parallel data that avoids overfitting for the forwarwd model, evident by a higher training PPL.

**MetaBT Flexibly Avoids both Overfitting and Underfitting.** We demonstrate two constrasting behaviors of MetaBT in Fig. 2 and Fig. 3. In Fig. 2, MetaBT generates pseudo-parallel data for the forward model to learn in WMT En-Fr. Since WMT En-Fr is large (40.8 million parallel sentences), the Transformer-Base forward model underfits. By "observing" the forward model's underfitting, perhaps via a low meta validation performance, the backward model generates the pseudo-parallel data that the forward model assigns a high probability, hence reducing the learning difficulty for the forward model. In contrast, Fig. 3 shows that for WMT En-De, the pseudo-parallel data generated by the backward model leads to a higher training loss for the forward model. Since WMT En-De has only 4.5 million parallel sentences which is about 10x smaller than WMT En-Fr, we suspect that MetaBT generates harder pseudo-parallel data for the backward model to avoid overfitting. In both cases, we have no control over the behaviors of MetaBT, and hence we suspect that MetaBT can appropriately adjusts its behavior depending on the forward model's learning state.

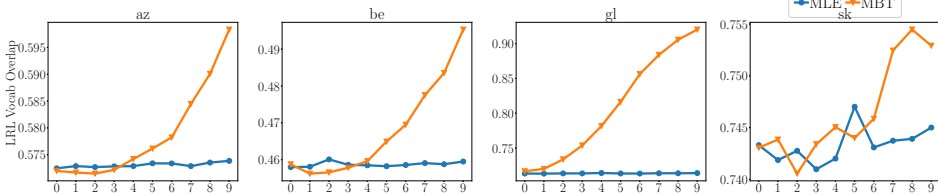

**Figure 4:** Percentage of words in the pseudo source sentences that are in the low-resource vocabulary throughout training. MetaBT learns to favor the sentences that are more similar to the data from the low-resource language.

**MetaBT Samples Pseudo-Parallel Data Closer to the Meta Validation Set.** After showing that MetaBT can affect the forward model's training in opposite ways, we now show that MetaBT actually tries to generate pseudo-parallel data that are closed to the meta validation data. Note that this is the expected behavior of MetaBT, since the ultimate objective is for the forward model to perform well on this meta validation set. We focus on the multilingual setting because this setting highlights the vast difference between the parallel data and the meta validation data. In particular, recall from § 4 that in order to translate a low-resource language $S$ into language $T$, we use extra data from a language $S'$ which is related to $S$ but which has abundant parallel data $S'$-$T$. Meanwhile, the meta validation set only consists of parallel sentences in $S$-$T$.

In Fig. 4, we group the sampled sentences throughout the forward model's training into 10 bins based on the training steps that they are generated, and plot the percentage of words in the pseudo source sentences that are from the vocabulary of $S$ for each bin. As seen from the figure, MetaBT keeps increasing the vocabulary coverage throughout training, indicating that it favors the sentences that are more similar to the meta validation data, which are from the low-resource language $S$.

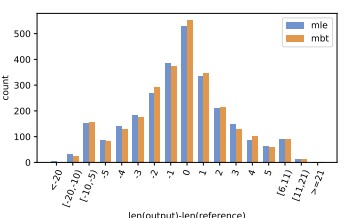

**Figure 5:** Histogram of differences in length between the reference and system outputs. MLE-trained BT tends to generate slightly more outputs with lengths that greatly differ from the reference.

**Qualitative Analysis: MetaBT Generates Fewer Pathological Outputs.** In Fig. 5, we plot the histogram of length differences between the reference sentences and the translations of MetaBT and by our baseline MLE on WMT En-De. We observe a consistent trend of the MLE baseline to generate more sentences with pathological length differences, i.e. more than $\pm 5$-10 words different from the reference's lengths. One such example is illustrated in Tab. 2 in § A.4. We suspect that this happens for MLE because while sampling-based back-translation increases diversity of the outputs and aids overall forward performance, it will still sometimes generate extremely bad pseudo-parallel examples. Forward models that learn from these bad inputs will sometimes produces translations that are completely incorrect, for example being too short or too long, causing the trends in Fig. 5. MetaBT suffers less from this problem because the backward model continues training to improve the forward model's dev set performance.

## 6  RELATED WORK

Our work is related to methods that leverage monolingual data either on the source side (He et al., 2020) or on the target side (Sennrich et al., 2016; Edunov et al., 2018) to improve the final translation quality. Going beyond vanilla BT, IterativeBT (Hoang et al., 2018) trains multiple rounds of backward and forward models and observe further improvement. While MetaBT cannot push the backward model's quality as well, MetaBT is also much cheaper than multiple training rounds of IterativeBT. DualNMT (Xia et al., 2016) jointly optimizes the backward model with the forward model, but relies on indirect indicators, leading to weak performances as we showed in § 5.4.

As MetaBT essentially learns to generate pseudo-parallel data for effective training, MetaBT is a natural extensions of many methods that learn to re-weight or to select extra data for training. For example, Soto et al. (2020) and Dou et al. (2020) select back-translated data from different systems using heuristic, while Wang & Neubig (2019); Lin et al. (2019); Wang et al. (2019b; 2020) select the multilingual data that is most helpful for a forward model. We find the relationship between MetaBT and these methods analogous to the relationship between sampling from a distribution and computing the distribution's density.

The meta-learning technique in our method has also been applied to other tasks, such as: learning initialization points (Finn et al., 2017; Gu et al., 2018), designing architectures (Liu et al., 2019), generating synthetic input images Such et al. (2019), and pseudo labeling (Pham et al., 2020).

## 7  LIMITATION, FUTURE WORK, AND CONCLUSION

We propose Meta Back-Translation (MetaBT), an algorithm that learns to adjust a back-translation model to generate data that are most effective for the training of the forward model. Our experiments show that MetaBT outperforms strong existing methods on both a standard NMT setting and a multilingual setting.

As discussed in § 5.3 the large memory footprint is a current weakness that makes it impossible to apply MetaBT to larger models. However, the resulting Transformer-Base model trained by MetaBT still outperform Transformer-Large models trained in the standard settings. Since the smaller Transformer-Base model are cheaper to deploy, MetaBT still has its values. In the future, we expect this memory limitation will be lifted, e.g. when better technology, such as automated model parallelism (Lepikhin et al., 2020) or more powerful accelerators, become available. When that happens, MetaBT's will better realize its potential.

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
