# OpenReview forum: "Meta Back-Translation"
_ICLR.cc/2021/Conference — ICLR 2021 Poster_

### Official Review · AnonReviewer4 · 2020-10-26
**interesting idea; concerns about evaluation**

**Rating:** 6
**Confidence:** 4

**Review:**

summary:

This paper applies techniques from meta-learning to derive and end-to-end update rule for a workflow involving backtranslation, specificically maximizing translation performance of the forward model, while updating the backward model to produce backtranslations that are maximally useful to improve the forward model's quality (as measured on a meta validation set). The approach is evaluated on WMT EN-DE and EN-FR and compared against a simple sampling strategy for backtranslation, and dual learning. In addition, the paper considers a multilingual setup where the translation direction is low-resource, and the initial backtranslation model is trained on a mix of parallel data from the language pair of interest, as well as auxiliary data with a high-resource, related source language.

strengths:

+ the idea of end-to-end optimization of the backward model to maximally benefit training of the forward model is novel and interesting.

+ the mathematical derivation of the objective function is sound.

weaknesses:

- the empirical evaluation is not very convincing. Important information is missing, baselines are inexplicably weak, and some other simple baselines are missing. Specifically:

 -- what data sets do you use for meta validation? Is it one (or several) of the newstest sets? Since you're actually learning gradients on this data set, rather than just using it for early stopping or hyperparameter choice, I think you should consider using that data directly for training as one of the baselines. For example, Oravecz et al. (2019) report a 1.5-2 BLEU gain from fine-tuning on newstest2008-2017 for EN-DE in their submission to WMT.

 -- for the Edunov et al. 2018 baseline (sampling for back-translation), there is a gap of >6 BLEU between the best result they report (35 BLEU) and yours (28.73 BLEU). Some of this may be explainable by the fact that you use Transformer base rather than Transformer large, but even relatively speaking, Edunov et al. (2018) observe an improvement of ~4 BLEU with BT, while this paper only reports ~2. Authors discuss memory limitations as the reason why the did not train on Transformer Big, but they could conceivably use gradient accumulation to enable training even on limited hardware. Other reasons for the score discrepancy deserve discussion (for example, are BLEU scores reported tokenized or not?)

 -- please provide more detail how the high-resource data was used for the various baselines. You mention that you follow settings from previous work (Neubig & Hu, 2018; Wang et al., 2019a), but these actually use different techniques. I gather that you use some type (which?) of transfer learning for the forward models. Do you also apply these techniques to the backward models, and if so, is the initial model for meta back-translation initialized differently?

 -- the dual learning baseline bears some conceptual similarity to the proposed objective in that both backward and forward model are continuously improved. I'm surprised to see that it leads to worse performance than the baseline trained on parallel data only, but there is too little information to build trust in this result. Which language model was used for the additional rewards? What beam size (or sample size) was used for the backtranslation? What monolingual data was used, only target-languge data or also source-language data? If the latter, which data was used?

- I find the title problematic. I can see how it was coined as a combination of meta-learning and backtranslation, but it's misleading to a reader who doesn't know this intention. If we take "meta" to roughly express some self-referentiality (meta-learning is learning to learn; a meta-analysis is an analysis of analyses), is meta-backtranslation the backtranslation of backtranslations?

recommendation:

Overall, I vote for rejection. I like the core idea and could see this work being published eventually, but feel that the empirical evaluation needs to be strengthened before I would recommend acceptance. There are open questions about the evaluation setup, and if authors can answer these questions, this will let me better judge the empirical rigour.

typos:

mulltilingual -> multilingual


**post-response update**:

some of my concerns about the evaluation were resolved in the author response (for example regarding the meta-validation datasets) , and I have slightly increased my rating.

---

> ### Author Response · Authors · 2020-11-13
> **Response to reviewer 4**
>
> We thank R4 for raising the detailed review, and clarifications about the missing details in the paper. Please find the answers to your questions below. We will include all these missing details in the future revision of our paper.
>
> **[Meta-validation datasets]** We remove 3,000 pairs of parallel training data from WMT EnDe / EnFr training datasets to use as the meta-validation set. For our baselines, we do not withhold these 3,000 pairs of parallel sentences, and so our baselines already use this data. This makes the comparison fair, as R4 suggested.
>
> **[Comments on Results for SamplingBT]** Our BLEU scores are calculated tokenized using moses tokenizer; we will clarify this in the paper. We suspect that the gap between NoBT and SampleBT baseline (MLE in the paper) is smaller than observed by Edunov et al. (2018) because of the small capacity of the Transformer-Base model. Perhaps a larger model is required to fit the 260M sentences in the German monolingual dataset that we and Edunov et al. (2018) both use.
>
> Another piece of evidence in our paper that supports this theory about small capacity is the fact that this improvement becomes smaller for our WMT EnFr experiments. Since the WMT EnFr dataset is larger than WMT EnDe, our small model becomes more underfit.
>
> Using small models does not invalidate our results, as we fairly use the same setting for all of our baselines. That said, we have acknowledged in the paper that somewhat larger memory footprint is currently a downside of MetaBT. In the future, we hope to alleviate this issue with better techniques, and it is also likely that the development of better hardware will reduce the issues somewhat as well.
>
> **[Use Gradient Accumulation to Train Larger Models]** Thank you for the insightful suggestion. In fact, we have thought about gradient accumulation when working on the paper. That said, we found that gradient accumulation is not trivial to use together with auto-differentiation to update the backward model, but we will certainly consider it for future experiments.
>
> Specifically, to accumulate gradients from multiple steps, in our framework TensorFlow, we need to use a tf.while_loop and need to aggregate the intermediate gradients in a memory block. Auto-differentiation computations apparently cannot propagate through this aggregation process, resulting in a None gradient for the backward model. A more viable approach for us would be model parallelism, but the technique is still not quite ready at the moment.
>
> **[More details on the Neubig & Hu. (2018) and Wang et al. (2019a)]** We are sorry about the confusion here. We mostly follow Neubig & Hu (2018)’s most simple method (the “Bi” method) of simply using the concatenation of low-resource language and its most related high-resource language for training both the forward and the backward model. We will clarify this in a revised version of the paper.
>
> **[Comments on DualNMT]** We were surprised that DualNMT’s results ended up being very weak too. To provide more details:
> - For WMT EnDe/EnFr, we use a separate Language Model (LM) for each language English, German, and French. These LMs are 6-layered Transformer-Base models, similar to the encoder of our translation Transformers. Each LM is pre-trained on the corresponding language’s monolingual data, for which we use the Newscrawl data between 2012-2018. All LMs use the same SentencePiece tokenization with the translation models (note that this means that we have different English LMs for WMT EnDe and WMT EnFr). During DualNMT, we use a sample size of 1, due to memory constraints.
> - We found DualNMT to be very slow and memory inefficient. In particular, DualNMT requires that we store in memory two translation models (one for each direction) plus two LMs (one for each language). The interactions between these models also require their intermediate computations to be kept in memory. As a result, we needed to halve the global batch size compared to the other methods.
> - DualNMT can perhaps be improved if we increase the LMs’ capacity. We suspect so, because we observe that our LMs are still underfitting the monolingual training data (their training PPL is in the ballpark of 50). However, larger LMs will not fit in memory together with two translation models.
> - For our multilingual low-resource experiments, we use the same BT sampling setup for both DualNMT and MetaBT to ensure the fairness of comparison. The LMs are 6-layered Transformer-Base models, and they are pretrained on the monolingual data from the low-resource language. We suspect that DualNMT does not perform well in this setting because the LMs are not trained on enough data.
>
> **[Title]** We thank R4 for understanding the intentions behind our title and method name. For better clarity, in our revision, we will extend the title to better reflect MetaBT’s intention. For example: “Meta Back-Translation: Generating Effective Pseudo-Parallel data with Back-Translation and Meta-Learning”.

---

### Official Review · AnonReviewer3 · 2020-10-28
**Missing some baselines**

**Rating:** 7
**Confidence:** 4

**Review:**

The paper proposed an interesting approach for back-translation. The idea is to update both forward model and backward models during training. The forward model is updated in a standard way using synthetic samples generated from the backward model. The backward model is updated using gradient of the forward model on meta-validation dataset (i.e., parallel data). Evaluation shows that Meta BT performs better than offline BT ((Edunov et al., 2018)  in both multilingual setup and standard  en-fr, en-de translation direction. In general, I find the approach is nice.
While the paper compared Meta-BT with offline BT (i.e., MLE as mentioned in the paper) and DualNMT, I think these two baselines are not sufficient to verify the claims made by the paper.

The paper claims that MetaBT allows to update the backward model unlike offline-BT and avoids expense of multiple iterations in iterative BT. Note that back-translation can be done on-the-fly (Artetxe et al., 2018, Conneau and Lample, 2019). Online back translation allows updating both forward and backward models using monolingual data during training. Thus, I think online BT should be a baseline for appropriate comparison. While MetaBT avoids multiple iterations of iterative BT, the evidence is not provided in the paper in terms of training time for MetaBT. The additional complexity of MetaBT lies in the update of the backward model per mini-batch and the computation of the Jacobian-vector products. As the author already mentioned that MetaBT has a large memory footprint, thus it’s slower to do one update in MetaBT. I wonder how MetaBT performs in comparison with iterative BT with 2 iterations with respect to BLEU scores and training time.

Can the authors provide similar plots in Figure 2 for En-De and in Figure 3 for En-Fr? With respect to MetaBT avoids overfitting. I think it would be nice to have some analysis on the samples generated by the backward model. In comparison to offline BT, does the backward model in MetaBT generate more diverse output?

With respect to the presentation of the paper, I think Figure 1 is a bit confusing to read. I was hoping to get the main idea from Figure 1 but it didn’t help at all.




**References**

Mikel Artetxe, Gorka Labaka, Eneko Agirre, Kyunghyun Cho. Unsupervised Neural Machine Translation. ICLR 2018.
Guillaume Lample, Alexis Conneau. Cross-lingual Language Model Pretraining. NeurIPS 2019


**Post-response update**
Thanks authors for extra effort on semi-supervised experiments. I decided to increase the score to 6.

---

> ### Author Response · Authors · 2020-11-13
> **Response to reviewer 3**
>
> We thank R3 for the comments and the suggestions for the baselines.
>
> **[Comparing to On-the-fly Back-translation]** Of the two baselines R3 suggested, we find that:
>
> - Cross-lingual Language Model Pretraining (CLM; Lample and Conneau, 2019) uses a pre-training step which is not used in MetaBT. Pre-training is certainly a reasonable method for using available data, but it is different in nature than back-translation methods such as MetaBT, so we decided to focus our experimental comparison on the most related techniques (which are also widely used in state-of-the-art translation systems). We also hypothesize that due to the different nature of these techniques the gains from them are likely to stack to some extent. Thus, we did not compare against CLM.
> - Unsupervised Neural Machine Translation (UNMT; Artetxe et al., 2018). We are implementing UNMT to compare with MetaBT on WMT’14 EnDe and EnFr, using the same Transformer-Base model and training data with MetaBT. We are working on running this experiment, and will try to update you with the results in a few days if the experiments finish within the time frame for authors/reviewers discussion (apologies if they do not -- the time frame is a bit tight).
>
> **[Training time]** Using 128 TPUv2 with the Transformer-Base model, our implementation MetaBT takes about 20 hours to train from scratch to convergence. In comparison, on the same hardware, the same model takes 18 hours to train on WMT-EnDe + monolingual BT (it takes 4 hours to train on only the WMT EnDe parallel data, but obviously the results are much worse). Therefore, as we wrote in the paper, MetaBT takes a bit longer than one iteration of back-translation, but is faster than two iterations of iterative back-translation.
>
> **[Extra Plots about MetaBT’s Behaviors on WMT EnDe & EnFr]** We provide the extra plots here. We will add them to the final version of the paper. Please find the figures here:
> * Figure 2 for en-de: https://pasteboard.co/JA1P6uz.png
> * Figure 3 for en-fr: https://pasteboard.co/JA1Q0e0.png and https://pasteboard.co/JA1Qi4C.png
>
> **[Does the backward model in MetaBT generate more diverse outputs?]** Yes. We have presented this analysis for our multilingual experiments in Fig. 4. The figure shows that MetaBT does learn to generate outputs that have more words from the low-resource language. These outputs are more diverse than that of the vanilla backward model, which mostly generates sentences from the high-resource language.
>
> *References*
>
> Mikel Artetxe, Gorka Labaka, Eneko Agirre, Kyunghyun Cho. Unsupervised Neural Machine Translation. ICLR 2018.
>
> Guillaume Lample, Alexis Conneau. Cross-lingual Language Model Pretraining. NeurIPS 2019.

---

> ### Author Response · Authors · 2020-11-17
> **Comparison with Unsupervised NMT**
>
> We have finished our experiments with UNMT. In particular, we trained a Transformer-Base model to translate En->De. We use the Semi-supervised mode of UNMT which was described at the end of Page 7 in Artetxe et al., (2018). We let the model learn on all parallel data in WMT’14 En-De. Additionally, we use the same amount of monolingual data in English for the denoising objective and the on-the-fly back-translation objective.
>
> We attain a BLEU score of 28.76 for WMT’14 En->De which is in the same ballpark with back-translation using sampling (SampleBT, 28.73 from Table 1 in our paper).
>
> Most importantly, this indicates that MetaBT achieves +1.63 BLEU higher than UNMT, even though UNMT uses monolingual English data while MetaBT does not. We will continue to analyze this result going forward towards the final version of the paper, but we believe that at least for now our result demonstrates MetaBT, whose sole purpose is to improve the quality of the forward translation model En->De, is being successful in this endeavor.
>
> There are also a few auxiliary negative points about UNMT compared to MetaBT:
> * First, UNMT needs 2 rounds of back-translation, 2 rounds of denoising, and 1 round of supervised training, specifically:
> noisy En **-> En**
> noisy De **-> De**
> mono En -> infer De **-> mono En**
> mono De -> infer En **-> mono De**
> parallel En **-> parallel De**
> All the **bolded arrows** are trained. Compared to UNMT, MetaBT only has:
> parallel En **-> parallel De**
> mono De -> **infer En -> mono De**
>
> * Second, we measured the step time of two methods and found that for the same batch size, UNMT is 1.4x slower than MetaBT, even though MetaBT keeps two models in memory and has to perform two back-propagation passes.

---

> > ### Comment · AnonReviewer3 · 2020-11-24
> > **double check**
> >
> > Dear authors,
> >
> > I would like to double check with the on-the-fly-BT experiment. The motivation for on-the-fly-BT is that one can update both forward (i.e., En->De) and backward (i.e., De->En) models during training. In this case, both forward and backward are first trained on parallel data. Then a mini-batch of _mono De_ is drawn, the backward translates it to _infer En_ and the forward model updates with (_infer En_, _mono De_). The next mini-batch is _mono En_, the forward model translates it to _infer De_ , and the backward model is updated with (_infer De_, _mono En_). Both forward and backward models are updated alternately.
> >
> > I wonder if this is the right way to read the  **bolded arrows** in your above comments?

---

> > > ### Author Response · Authors · 2020-11-24
> > > **Your interpretations are correct**
> > >
> > > Your interpretations of the **bolded arrows** are all correct. We are very grateful that you engage in discussing with us, even on the details :-)
> > >
> > > Just to make everything crystal clear, in your comment you wrote *"both forward and backward are **first** trained on parallel data"*. The word *"**first**"* should refer to the updating order *in a training iteration*, and not that the forward and backward models are first pre-trained to convergence on parallel data and then are trained on monolingual data. This follows the UNMT paper's descriptions. Specifically, at the end of Page 7: *"their training **alternates** between denoising, backtranslation and, additionally, maximizing the translation probability of these parallel sentences"*

---

> > > > ### Comment · AnonReviewer3 · 2020-11-24
> > > > **Thanks for elaboration**
> > > >
> > > > Thanks for the elaboration! When I referred to on-the-fly BT I didn't mean to use it in the same way UNMT of Artetxe et. al.,  used. I was wondering if you **pre-train** both forward and backward first then the both models can generate better back-translated data, which in turn, can be used to update both models. This perhaps could be a stronger baseline.
> > > >
> > > > Nevertheless, I think there is a merit  in the proposed method. I decided to change my score to 7.

---

> > > > > ### Author Response · Authors · 2020-11-24
> > > > > **Let's try pre-training**
> > > > >
> > > > > Thank you for the positive comment and for increasing the score. It means a lot to us.
> > > > >
> > > > > We will take some time to try your suggestion on pre-training and continual training. Specifically, we'll pre-train a shared encoder and a decoder into both languages until convergence, and then we will continue to train on the denoising objective and the two-round back-translation objective.
> > > > >
> > > > > Please note that pre-training a shared encoder translation model in both directions and then continuing to train the converged model with both the denoising objective and the two-round back-translation objective will lead to a significantly slower training time compared to UNMT, which is already slower than MetaBT.
> > > > >
> > > > > We will update you about the result when we get it.

---

> > > > > > ### Comment · AnonReviewer3 · 2020-11-24
> > > > > > **Sound exciting!**
> > > > > >
> > > > > > Thanks you for trying this experiment. I'm curious to see the results.
> > > > > >
> > > > > > Artetxe et. al, use denoising objective since they assume a small amount of parallel data (10K - 100K) available. I think it's not necessary to use denoising objective in this experiment if you use full WMT14 parallel data. Using denoising objective after both forward and backward models converge might hurt both models since they will learn to copy source to target.
> > > > > >
> > > > > > I think it's sufficient to train forward and backward models till convergence and then use on-the-fly-BT to update both.

---

### Official Review · AnonReviewer2 · 2020-10-28

**Rating:** 7
**Confidence:** 5

**Review:**

The paper describes a method to improve NMT training with backtranslation.

Rather than using a fixed t->s model to translate target monolingual data in order to augment the training set for the s->t model, the proposed approach first pretrains the t->s model as usual then jointly trains it with the forward s->t model using a meta-learning approach: the s->t model is trained on the syntetic backtranslated data and a "meta-validation" loss is computed on a paralled dataset, which is used to update the t->s model using REINFORCE.
The approach is similar to the DualNMT model by Xia et al, but rather than updating on monolingual data based on LM and reconstruction scores, it uses a reward based on the cross-entropy on parallel data.
The paper also proposes a way to adapt this method to a multi-lingual setting.

Experiments are performed on WMT-14 En->De and En->Fr, and on 4 IWSLT-2018 language pairs. The authors report small but consistent improvements. Additional analyses are also reported.

Overall the method seems valid, although it is described at a very high level and no code release is mentioned. In my experience successfully implementing RL-based method is strongly dependent on getting hyperparameters and implementation details right, so it could be hard to reproduce this work without the code or a more detailed description. Also it's not entirely clear what is being used as "meta-validation" data here, I suppose it's all the parallel training data, but the paper doesn't make it clear.

Minor issues: the "Tagged backtranslation" paper by Caswell et al. 2019 contrasts the claim that improvements with sampling backtranslation are due to increased diversity. It should be referenced as relevant work. The Xia et al., 2016 Dual NMT paper is referenced multiple times in the text but not in the bibliography section

---

> ### Author Response · Authors · 2020-11-13
> **Response to reviewer 2**
>
> We thank R2 for the overall positive comments and suggestions.
>
> **[Meta-validation datasets]** We removed 3,000 pairs of parallel training data from WMT EnDe / EnFr training datasets to use as the meta-validation set. Note that for our baselines, we do not withhold these 3,000 pairs of parallel sentences, and so our baselines already use this data. This makes the comparison fair, as R4 suggested.
>
> We thank the reviewer for pointing out the extra references. We will include them as well as the Xia et al. paper in the related work.

---

### Official Review · AnonReviewer1 · 2020-10-31
**An interesting application of meta-learning in NMT, but with an unclear motivation**

**Rating:** 6
**Confidence:** 4

**Review:**

The paper presents an extension of the back-translation method which provides a means of leveraging monolingual data in NMT where the quality of the data generated by the back-translation model is controlled through a meta learning regime that trains the BT model jointly with the actual translation model.
The method is an interesting application of meta-learning to NMT and worth seeing the results. It is a well-written paper with a sound description of the method and evaluation. The motivation is the main weakness and a better discussion and comparison to related work would help clarify the applicability of the method.

Comments
- The motivation for the method is supported by two claims, the first one related to the upper bound on quality of the BT model trained on the parallel data, and the second on the quality of the pseudo-parallel data generated by the BT model used then to train the actual NMT model together with the original parallel data. There is on the other hand not any empirical support in either claim that these create a weakness, thus, grounding the motivation for the paper.
In the former case, the method does not really modify in any matter the quality of the original parallel data, hence the discussion is irrelevant. In the latter case, especially, when using the pseudo-parallel data, there are many factors not discussed that affect the quality of the pseudo-parallel data generated by the BT model other the diversity, such as source domain drift and eg. generation of translationese. It is indeed limited the literature on this topic although doesn't validate that any claim should be supported either by references or within the study included in the paper. The second reference in this context is referred to as slow and expensive but isn't the meta learning also making the model much slower and more costly in a similar sense?
- Can the authors analyze or discuss how well the backward model is improving and how does it distinguish from the previous methods?

---

> ### Author Response · Authors · 2020-11-13
> **Response to reviewer 1**
>
> We thank R1 for the overall positive comments and suggestions.
>
> **[Clarity of motivation]** To reiterate the introduction somewhat, the main goal of our method is to directly optimize the backward model so that it produces data that are most effective to train the forward model. Because of this, it is a bit of a mischaracterization that our method is attempting to improve the quality of the backward model itself, but rather we are trying to improve the positive effect that these translations have on improving training of the forward model. Because of this, the only true measure of whether our model has succeeded is if the forward translation quality improves, and our experimental results demonstrate that this is the case. This is a new way of thinking about back translation and proposing this new way of thinking about back translation is one of the major contributions of this paper. As a corollary to this, we are also not aware of any relevant citations that assess back-translation quality from this standpoint, but if you can point out any particularly relevant ones we would be happy to cite them.
>
> **[Analysis of how the BT model is improved]** While forward translation model improvements are the only true indicator of whether our proposed method has succeeded, Figure 4 does show one piece of analysis: MetaBT learns to generate sentences that are closer to the low-resource language, although the initial backward model is trained to mostly generate the related high-resource language. This could also be viewed as one variety of “domain shift” as mentioned by in the review, if we view the two different input languages as different domains.

---

### Author Response · Authors · 2020-11-22
**We'd like to address any more questions about the paper!**

Thanks the reviewers for giving us a lot of good feedback. We hope the rebuttal addressed your concerns about our paper, and we would love to answer and discuss any other questions you have!

---

### Decision · Program_Chairs · 2021-01-07
**Final Decision**

**Decision:**

Accept (Poster)

**Comment:**

This paper proposes a meta-learning-based technique to learn how to back-translate (generate a synthetic source-language translation of an observed target-language sentence) for the purpose of better optimising a source-to-target translation model.

The approach is an interesting novel angle to jointly training the translation model and the back-translation component. Compared to techniques like UNMT and DualNMT, the approach offers reduced training time and a simpler formulation with fewer trainable components (and fewer hyperparameters).

During the discussion phase the authors provided additional insight, clarifications, and results that improved our perception of the paper. I would personally appreciate if the authors would update their paper with the clarifications they made to points raised by R2, R3, and R4, especially on the details about meta-validation, the discussion about memory footprint, and the additional results on UNMT (and variants).